# TREM2-Deficient Microglia Attenuate Tau Spreading In Vivo

**DOI:** 10.3390/cells12121597

**Published:** 2023-06-10

**Authors:** Audrey Lee-Gosselin, Nur Jury-Garfe, Yanwen You, Luke Dabin, Disha Soni, Sayan Dutta, Jean-Christophe Rochet, Jungsu Kim, Adrian L. Oblak, Cristian A. Lasagna-Reeves

**Affiliations:** 1Stark Neurosciences Research Institute, Indiana University School of Medicine, Indianapolis, IN 46202, USA; audrey.lee.gosselin@gmail.com (A.L.-G.); njurygar@iu.edu (N.J.-G.);; 2Department of Anatomy, Cell Biology & Physiology, Indiana University School of Medicine, Indianapolis, IN 46202, USA; 3Department of Medical & Molecular Genetics, Indiana University School of Medicine, Indianapolis, IN 46202, USA; 4Department of Radiology & Imaging Sciences, Indiana University School of Medicine, Indianapolis, IN 46202, USA; 5Department of Medicinal Chemistry and Molecular Pharmacology, Purdue University, West Lafayette, IN 47907, USA; 6Purdue Institute for Integrative Neuroscience, Purdue University, West Lafayette, IN 47907, USA; 7Center for Computational Biology and Bioinformatics, Indiana University School of Medicine, Indianapolis, IN 46202, USA

**Keywords:** Alzheimer’s disease, Trem2, microglia, tau propagation, tauopathy, neuroinflammation

## Abstract

The role of TREM2 in Alzheimer’s disease (AD) is not fully understood. Previous studies investigating the effect of TREM2 deletion on tauopathy mouse models without the contribution of b-amyloid have focused only on tau overexpression models. Herein, we investigated the effects of TREM2 deficiency on tau spreading using a mouse model in which endogenous tau is seeded to produce AD-like tau features. We found that Trem2^−/−^ mice exhibit attenuated tau pathology in multiple brain regions concomitant with a decreased microglial density. The neuroinflammatory profile in TREM2-deficient mice did not induce an activated inflammatory response to tau pathology. These findings suggest that reduced TREM2 signaling may alter the response of microglia to pathological tau aggregates, impairing their activation and decreasing their capacity to contribute to tau spreading. However, caution should be exercised when targeting TREM2 as a therapeutic entry point for AD until its involvement in tau aggregation and propagation is better understood.

## 1. Introduction

Coding alterations in TREM2, a transmembrane glycoprotein expressed on myeloid lineage cells that modulates innate immune function [1], dramatically increase the risk of developing late-onset Alzheimer’s disease (LOAD) [2,3]. TREM2 coding variants have also been associated with increased risk for amyotrophic lateral sclerosis (ALS) and frontotemporal-lobar dementia (FTLD) [4,5,6]. Several studies have aimed to dissect the effect of TREM2 on beta-amyloid (Ab) and tau aggregation in a diverse set of AD mouse models [7,8,9,10,11,12,13,14,15]. Conflicting results have been reported regarding amyloid plaque deposition and the effect of TREM2 loss [7,8,9,10,11], suggesting that the effect of TREM2 on Ab deposition may be dependent on age, sex, and the model utilized.

TREM2 loss-of-function studies have also yielded conflicting outcomes in the context of tau pathology. For instance, TREM2 ablation in the hTau model that expresses only human tau isoforms [16] exacerbates tau hyperphosphorylation and aggregation [12]. However, TREM2 ablation was protective and decreased synaptic impairment and neurodegenerative phenotypes [13] in the PS19 mouse model [17]. In another study that evaluated the effect of TREM2 loss-of-function in the PS19 model, a 50% decrease in TREM2 but not a total reduction enhanced tau phosphorylation and aggregation [18]. Furthermore, in the THY-Tau22 transgenic model that overexpresses human tau carrying the G272V and P301S mutations, it was shown that TREM2 deletion increases tau hyperphosphorylation and aggregation at advanced stages of pathology [15]. In a recently published study, TREM2 deletion was found to exacerbate tau pathology and promote brain atrophy in a mouse model that overexpressed human tau with the P301L mutation and human amyloid precursor protein (APP) but had no effect in a model that solely overexpressed mutant tau, suggesting that TREM2 function is neither protective nor detrimental in a pure tauopathy setting [14]. The notion that TREM2 influences tau pathology via amyloid accumulation is further supported by another study in which isolated pathological tau aggregates from human AD brains were injected into the brains of mice harboring beta-amyloid pathology [19]. TREM2 deletion increased the abundance of hyperphosphorylated tau dystrophic neurites surrounding amyloid plaques [19]. Collectively, these studies yielded disparate results regarding the effect of TREM2 on tau pathology and whether the effect is amyloid dependent or independent.

All previous in vivo studies evaluating the effect of TREM2 deletion on tauopathy mouse models without the contribution of b-amyloid have been conducted in tau overexpression models; thus, the effect of TREM2 ablation on tau propagation remains unclear. Here, we used a mouse model in which endogenous tau is seeded to produce AD-like tau features via sarkosyl-insoluble tau aggregates isolated from the frontal cortex of human AD brain tissue to specifically investigate the effects of TREM2 deficiency on tau spreading [20]. C57BL/6J (wild type (WT)) and Trem2^−/−^ (KO) mice were unilaterally injected with insoluble human tau aggregates. TREM2 deletion attenuated tau spreading, potentially due to a decrease in microglial function. The data presented here suggest that TREM2 deletion may be beneficial in diminishing tau propagation in a pure tauopathy setting.

## 2. Material and Methods

### 2.1. Animals

Trem2^−/−^ mice were obtained from The Jackson Laboratory (JAX; Strain #027197). Control C57BL/6J (WT: wild type) mice were purchased from JAX (Strain #000664). Mice were housed at the Indiana University School of Medicine (IUSM) and were maintained on a 12-h. light/dark cycle with food and water ad libitum in accordance with the Guide for the Care and Use of Laboratory Animals (National Institutes of Health, Bethesda, MD, USA). Mice were bred in-house. Intracranial injections of sarkosyl tau were administered to 3-month-old mice. At 9 months of age, the mice were anesthetized for transcardial perfusion with ice-cold phosphate buffered saline (PBS). Hemibrains were frozen or fixed (10% formalin) for the subsequent experiments. All experimental protocols and animal care were approved by the IUSM Institutional Animal Care and Use Committee (IACUC).

### 2.2. Purification of Tau from Human AD Brain Tissue

Human brain tissue was obtained from the National Centralized Repository for Alzheimer’s Disease and Related Dementias (NCRAD) (ncrad.iu.edu). Brain tissue for paired helical filament (PHF) tau extraction was obtained from an 86-year-old female diagnosed with Alzheimer’s disease (AD; Braak stage 6). The presence of tau in the temporal cortex was histologically confirmed using AT8 (1:500, MN1020, Invitrogen, Waltham, MA, USA) [20]. Briefly, 6 g of frontal cortical gray matter was homogenized in a high-salt buffer with 0.1% sarkosyl and 10% sucrose. After centrifugation and re-extraction of the pellet, the supernatants were pooled, and sarkosyl was added to a final concentration of 1%. After incubation and centrifugation, the pellets were washed and resuspended in PBS, sonicated, and then centrifuged again to collect the enriched pathological tau (AD-tau) in the supernatant.

### 2.3. Characterization of AD-Tau

To characterize the tau extracted, first, we quantified the total tau concentration using a V-PLEX Human Total Tau Kit (K151AE-1, MesoScale Discovery, Rockville, MD, USA), following the manufacturer’s protocol. Negative staining electron microscopy (EM) was used to confirm the presence of fibrils in the extracted tau. The sample was diluted 8-fold; then, 3 μL of the AD brain-extracted sample was added to a glow-discharged carbon-coated copper transmission electron microscopy (TEM) grid, which was subsequently incubated for 45 s [21]. The grid was washed with deionized water and stained with 3.5 μL of 1% (*w*/*v*) phosphotungstic acid (PTA) solution for 1 min. Grids were dried by blotting the excess PTA with Whatman filter paper and imaged using an FEI Tecnai T12 Transmission Electron Microscope operating at 80 kv. Gatan MSC794 digital micrograph software was used to capture the images. Length measurements were conducted using ImageJ. We also performed flow cytometric fluorescence resonance energy transfer (FRET) seeding assays as previously described [22] to evaluate the seeding capability of the AD-tau. Seeding was quantified by measuring the percentage of positive cells and the median fluorescent intensity (MFI) of the compensated BV510 channel. Integrated FRET density was calculated by multiplying the percentage of positive cells with the MFI.

### 2.4. Stereotaxic Surgery

Three-month-old WT and Trem2^−/−^ mice were deeply anesthetized via an intraperitoneal injection of tribromoethanol. The head was fixed between ear bars in a stereotaxic frame with a manipulator arm and microinjection pump (Angle Two, Leica Biosystems, Nußloch, Germany). The dorsal hippocampus and overlying cortex of one hemisphere were aseptically injected with human AD-tau using a 33-gauge Hamilton syringe (coordinated from bregma: anterior-posterior −2.5 mm, lateral 2 mm, and depth −2 mm and −0.85 mm from the skull). Brain extracts were injected into the hippocampus before the needle was pulled upward to the cortical injection site. A concentration of 1 μg/μL (2.5 μL) was injected at each site for a total volume of 5 μL (5 µg) per mouse.

### 2.5. Mouse Brain Lysates for Protein and RNA Preparation

Brains were homogenized at a 1:10 (*w*/*v*) ratio of brain tissue to 1X tris-buffered saline (TBS) containing a complete protease inhibitor cocktail (ThermoFisher Scientific, Waltham, MA, USA). Brains were homogenized using a beadbeater at 1400 rpm for 30 s. The ipsilateral and contralateral brain sides were kept separate. Lysates were separated into two aliquots for RNA preparation and protein preparation. For protein preparation, samples were centrifuged at 14,000 rpm for 15 min at 4 °C. The supernatant was collected, and the protein concentration was measured using a bicinchoninic acid (BCA) assay (ThermoFisher Scientific). For RNA preparation, TRIzol (Invitrogen) and chloroform were added, and the lysates were incubated for 15 min on ice. After centrifugation, the clear upper aqueous layer containing RNA was transferred to a new tube and an equivalent volume of 70% ethanol was added before continuing the purification with the PureLink™ RNA Mini Kit (Invitrogen).

### 2.6. Immunohistochemistry (IHC) and Immunofluorescence (IF)

Mouse brains were fixed in formalin and embedded in paraffin. The brains were then sliced coronally into 10 μm sections spanning from the midbrain to the frontal cortex. To ensure accurate comparisons between experimental conditions, equivalent brain sections were selected for every region analyzed, using the injection location identified by Nissl as a reference. Two sections per animal (Bregma −2.80, −2.18), capturing all affected brain structures, were used. The sections were deparaffinized in xylene, rehydrated with a series of ethanol concentrations (100% to 30%), and washed with deionized water. Heat-induced antigen retrieval high pH (4956-58, Invitrogen) was performed. The sections were then blocked with PBS containing 5% horse (IHC) or 10% (IF) goat serum and 0.1% Triton X-100 for 1 h at room temperature (RT) and subsequently incubated overnight at 4 °C with the following specific primary antibodies diluted in blocking solution: anti-AT8 1:200 (MN1020, ThermoFisher Scientific) (IHC), anti-IBA1 1:200 (019-19741, Fujifilm Wako Chemicals, Richmond, VA, USA), or anti-glial fibrillary acidic protein (GFAP) 1:300 (G3893, MilliporeSigma, Burlington, MA, USA) (IF). The next day, sections were washed 3 times in PBS and incubated with 1:500 biotinylated horse anti-mouse antibody (BA-200, Vector) for 1 h at RT (IHC) or with 1:300 goat anti-rabbit Alexa-Fluor-568 in blocking buffer for 90 min (IF). The Vectastain Elite ABC peroxidase kit (PK-6100, Vector Laboratories, Newark, CA, USA) was prepared 30 min in advance for immunohistochemistry (IHC) according to the manufacturer’s instructions. After the secondary antibody incubation, sections were incubated with the A + B solution for 30 min at RT. After 3 PBS washes, the DAB substrate was applied for 2 min for visualization, and Nissl was used for counterstaining. Slides were immersed in 95% and 100% ethanol and then in xylene before mounting with a coverslip. For IF, the following secondary antibody incubation sections were washed in PBS, mounted with Vectashield mounting medium with DAPI (SK-4100, Vector Laboratories), and sealed with nail polish. The images were captured using a Leica DM2500 LED microscope with a Leica DMC 4500 camera (IHC) or a Nikon A1-R laser-scanning confocal microscope coupled with Nikon AR software v.5.21.03 (IF).

### 2.7. Microscopy Image Analyses

Images were acquired using a 40X objective to quantify tau (AT8) pathology. The images were thresholded, and the percentage area was analyzed using ImageJ. For microglial and astrocytic parameter quantifications, images were taken using a 63X objective with a ×0.7 zoom and a 0.3-μm z-step on a Nikon A1-R laser-scanning confocal microscope. ImageJ and the Nikon NIS-Elements imaging software v.5.21.03 were used to analyze the images. The total number of cells in each image was quantified by counting DAPI-positive cells, and IBA1^+^/GFAP^+^ cells were manually counted. The glial branches were measured using cross-sectional measurements of 2–10 randomly skeletonized and selected cells per animal (*n* = 4–6) using *z*-stack images in ImageJ, as previously described [12].

### 2.8. NanoString

Gene expression was evaluated in a different cohort of mice using the NanoString mouse neuroinflammation panel on the nCounter MAX system. In short, 5 μL (50 ng) of RNA extracted from the ipsilateral whole hemisphere was incubated overnight at 65 °C in nuclease-free H_2_O with Reporter and Capture CodeSet probes. After hybridization, the samples were loaded into the prep station, and barcodes were counted on the nCounter Digital Analyzer. Data analysis was conducted using nSolver software v4.0. Normalized counts were used to further analyze the data using the NanoString Advance Analysis module or by transforming the data and importing it into R. The Limma v3.44.3 package was used to construct linear models of comparisons. Heatmaps were plotted using https://www.bioinformatics.com.cn/en (last accessed on 17 March 2023), a free online data analysis and visualization platform. The statistical significance levels for the comparison models and volcano plot generation were corrected for multiple comparisons using the Benjamini–Hochberg false discovery rate (FDR) method. Differentially expressed genes (DEGs) were those with a *p* < 0.05 and log_2_ fold change > ±0.58.

### 2.9. Statistical Analyses

The statistical significance of experiments involving two groups was assessed using an unpaired Student’s *t* test. A two-way analysis of variance (ANOVA) with Tukey’s multiple comparison correction was used to analyze 4 groups. *n* = 4–6 mice were used per experiment, including an equal number of females and males. Data are presented as the mean ± S.E.M. GraphPad Prism software v.9.5.0 (525) was used for all statistical analyses.

## 3. Results

### 3.1. Trem2^−/−^ Mice Show Attenuated Tau Pathology in Multiple Brain Regions

Abnormal human tau recruits and induces endogenous mouse tau accumulation and propagation in the brains of WT mice [20]. Here, 3-month-old WT and Trem2^−/−^ mice were inoculated with AD-tau to evaluate the role of TREM2 on tau spreading. Tau pathology was confirmed in the AD human brain used for tau extraction (Figure 1A), and the isolated AD-tau exhibited a fibril conformation (Figure 1B). Moreover, isolated AD-tau was transfected into tau RD P301S fluorescence resonance energy transfer (FRET) biosensor cells and demonstrated strong seeding activity (Figure 1C) in the quantification of the integrated FRET density via flow cytometry [22].

As in previous studies, AD-tau was unilaterally injected into the dorsal hippocampus and the cortex overlying the hippocampus [20]. Ipsilateral and contralateral brain regions anatomically connected to the injection sites (mammillary area (M), ventral hippocampus (VH), and retrosplenial cortex (RC)) were analyzed (Figure 2A). We confirmed the presence of pathological phospho-tau species six months post-injection using AT8 immunostaining (Figure 2B). Trem2^−/−^ mice demonstrated an overall reduction in tau inclusions (Figure 2C). No significant differences in AT8 staining were observed on the ipsilateral sides of Trem2^−/−^ and WT mice, possibly due to higher tau accumulation at the injection site. However, there was a significant reduction in AT8 staining in areas not directly adjacent to the injection site where the accumulation of pathological tau may be attributed to tau propagation, such as the mammillary area and the contralateral retrosplenial cortex.

### 3.2. TREM2 Deletion Attenuates the AD-Tau Propagation-Derived Increase in Microglial Density

Microglia are critical contributors to tau propagation [23]. Therefore, we investigated whether changes in microglia or astroglia were associated with the AT8 reduction in the mammillary area and the contralateral retrosplenial cortex. Immunofluorescence staining of the microglial activation marker IBA1 (Figure 3) was conducted in the five previously mentioned brain areas. First, AD-tau inoculation into WT mice increased the number of microglia in the mammillary area (Figure 3A,F). A trend (*p* = 0.0858) of increased microglia was observed in the contralateral retrosplenial cortex (Figure 3B,G) and the ipsilateral ventral hippocampus (Figure 3C,H). Notably, high levels of AT8 inclusions were detected in the mammillary area and retrosplenial cortex of these mice. Trem2^−/−^ mice injected with AD-tau showed significantly fewer microglia cells in the mammillary area (Figure 3A,F), contralateral retrosplenial cortex (Figure 3B,G), and ipsilateral ventral hippocampus (Figure 3C,H), as well as a trend (*p* = 0.066) towards a lower count in the ipsilateral retrosplenial cortex (Figure 3D,I). No differences in microglia numbers were found in the contralateral ventral hippocampus (Figure 3E,J) of Trem2^−/−^ versus WT mice injected with AD-tau. The microglial numbers did not differ between WT and Trem2^−/−^ mice injected with PBS (Figure 3F–J), suggesting that a reduction in TREM2 did not affect microglial numbers in nonpathological conditions. No differences in cell morphology (microglial branching) between WT and Trem2^−/−^ mice were observed in comparison with their respective controls (Figure 3F–J and Appendix A), suggesting that a lack of TREM2 alters microglial proliferation or survival but not reactivity in response to AD-tau propagation. The Pearson correlation analysis between AT8 values (Figure 2C) and the number of IBA1 positive cells (Figure 3F–J) suggested that mice with a lower level of phospho-Tau AT8 have a lower number of IBA1-positive microglia, specifically in the mammillary area (Appendix A).

We then assessed astrocytes using the astroglial activation marker GFAP to investigate whether other glial cells were involved in the response to tau pathology (Figure 4A). No statistically significant differences in the number of GFAP-positive astrocytes were found between groups in any of the previously examined areas (Figure 4B–D).

### 3.3. TREM2 Ablation Reduces the Neuroinflammatory Response Associated with Tau Propagation

Next, we characterized the neuroinflammatory response to tau propagation in Trem2^−/−^ mice using RNA extracted from whole-brain homogenates prepared from the ipsilateral hemisphere. NanoString analyses were performed using the neuroinflammation panel to assess the expression level of 770 genes, enabling a comprehensive assessment of 23 neuroinflammatory pathways (Appendix A). AD-tau-injected Trem2^−/−^ mice showed a lower expression of genes related to microglial function (*Tbc1d4*, *Nrm*, *Ltc4s*, *CD86*, *C3ar1*, *Slamf9*, *Mmp12*), the adaptative immune response (*Egfr*, *Ptprc*, *Ago4*, *CD19*, *Was*, *Fcrlb*), inflammatory signaling (*lkbke*), and cytokine signaling (*Osgin1*, *Egfr*, *CD70*) than AD-tau-injected WT mice (Figure 5A, pink dots). Additionally, genes related to microglial function and inflammatory signaling (*Ccl4*, *Cxcl0*, *Tlr2*, *Ccl3*, and *Rsad2*) were downregulated in Trem2^−/−^ mice injected with PBS compared with the respective controls (Figure 5A, green dots on left side of graph). In addition, we analyzed pathway annotations’ scores (Figure 5B), which consolidate information from multiple genes belonging to a specific pathway into a single score. This approach allowed us to evaluate the overall signature scores obtained using NanoString, and we observed significant alterations in several group comparisons. First, as expected, WT mice injected with AD-tau (WT AD-tau) showed a higher inflammatory profile than WT mice injected with PBS (WT-PBS). Second, Trem2^−/−^ mice injected with PBS (Trem2^−/−^ PBS) showed a lower neuroinflammatory profile than WT-PBS mice. These data suggest that TREM2 depletion alone modulates most of the pathways evaluated using NanoString. Third, Trem2^−/−^ mice injected with AD-tau (Trem2^−/−^ AD-tau) showed decreased microglial function and inflammatory signaling in comparison with WT AD-tau mice. These data suggest that TREM2 depletion dampens the effects of AD-tau injection, elevating the inflammatory response to near-control levels (WT-PBS Figure 5B and Appendix A).

Differential gene expression analysis revealed changes in genes related to microglia function in Trem2^−/−^ AD-tau mice, whereas Trem2^−/−^ PBS mice exhibited alterations in genes related to both microglial function and inflammatory signaling. These findings suggest that microglial function is impacted by Trem2 depletion, with specific genes becoming downregulated in response to AD-tau, while inflammatory signaling is altered primarily due to TREM2^−/−^ deficiency. We next investigated the individual scores for each of the four groups in both NanoString annotations. The microglial function score (Figure 6A) was significantly higher in WT AD-tau mice than in WT-PBS mice; however, no differences were observed in any of the comparisons with Trem2^−/−^ mice. However, the individual normalized gene expression levels of three microglial function-related genes revealed that *C3ar1* (Figure 6B), *Psmb8* (Figure 6C), and *Mmp12* (Figure 6D) were downregulated in Trem2^−/−^ AD-tau mice versus WT AD-tau mice. Notably, in a previous study, *C3ar1* (C3a receptor) deletion rescued tau pathology and attenuated neuroinflammation, synaptic deficits, and neurodegeneration in the PS19 tauopathy mouse model [24]. Conversely, the proteasome subunit (*Psmb8*) and matrix metalloproteinase-12 (*Mmp12*) are associated with detrimental roles in the brain [25,26]. The inflammatory signaling score also differed significantly in WT-PBS versus Trem2^−/−^ PBS mice (Figure 6E), suggesting that the annotation score differences arise from TREM2 loss. The normalized gene expression of three representative genes (*Rsad2*, *Crebbp*, and *Fcer1g*) in this pathway are shown in Figure 6F–H. A gene heatmap of each selected pathway is shown in Figure 6I,J. Distinct cluster patterns were observed across the four groups, revealing the upregulation and downregulation of specific gene groups. Interestingly, there was a trend observed in the astrocyte function score, which appeared lower in the Trem2^−/−^ AD-tau mice compared to the WT AD-tau mice, suggesting that Trem2 deficiency could influence how astrocytes react to pathological tau propagation. While this difference was not significant (Appendix A), the analysis of the individual normalized gene expression levels of astrocyte function-related genes revealed that *Osgin1* (Appendix A) and *Cd109* (Appendix A) were downregulated in Trem2^−/−^ AD-tau versus WT AD-tau mice. Furthermore, *B3gnt5* (Appendix A) was downregulated in Trem2^−/−^ PBS versus WT PBS mice. Notably, *Cd109* and *B3gnt5* are known markers of neuroprotective astrocytes [27,28], and Osgin1 protects astrocytes against oxidative insults [29]. Differential gene expression data for every gene included in the astrocyte function score are shown in Appendix A. Overall, these data suggest that TREM2 deficiency in the context of tau propagation could affect astrocyte function by decreasing the expression of known anti-inflammatory and neuroprotective factors. Nevertheless, further studies are necessary to determine the specific role of astrocytes in tau spreading and the possible involvement of TREM2 in this process.

## 4. Discussion

In the present study, we injected human AD-tau into WT and Trem2^−/−^ mouse brains and demonstrated that TREM2 deficiency reduces tau propagation in vivo. Our histological analysis showed increased IBA1-positive microglia numbers in WT mice injected with AD-Tau. However, this increase was not observed in Trem2^−/−^ mice injected with AD-tau in areas where tau spreading was significantly decreased. On one hand, the transcriptomic-based analysis demonstrated that microglial function is enhanced in AD-tau WT mice but not in AD-tau Trem2^−/−^ mice in comparison with that in PBS-injected controls, suggesting that TREM2 is involved in the microglial response associated with tau propagation throughout the brain. On the other hand, the transcriptomic analysis also demonstrated that the inflammatory response is impaired in Trem2^−/−^ mice injected with PBS, suggesting that the effect of TREM2 deficiency on the inflammatory response is not dependent on the presence of tau pathology. Overall, this finding suggests that TREM2-deficient mice demonstrate impaired signaling pathways independent of tau pathology that still influence tau propagation and other signaling pathways that are directly impaired due to the presence of tau pathology and subsequent propagation.

Regarding the role of TREM2 in AD, recent in vivo studies suggest that TREM2 may be protective or damaging depending on the pathological stage [19,30,31,32]. There is a complicated body of literature surrounding the role of TREM2 in tau pathology. For instance, TREM2 ablation significantly increased tau pathology in the hTau mouse model expressing all six isoforms of human tau [12]. Similarly, downregulating TREM2 levels in adult transgenic PS19 mice expressing human tau harboring the P301S mutation increased tau hyperphosphorylation and microgliosis [33]. Conversely, in another study, total deletion of TREM2 did not affect tau pathology in PS19 mice crossed with Trem2 KO mice; however, brain atrophy and microgliosis were reduced [13]. Sayed and colleagues also observed that a complete deficiency in TREM2 protected PS19 mice against tau-mediated microgliosis and brain atrophy [18]. Interestingly, TREM2 haploinsufficiency (50% reduction in TREM2 levels) increased tau pathology [18]. Thus, the effect of TREM2 on tau pathology likely depends on the overall expression levels of TREM2 and the stage and type of pathology, which is model-dependent. These conflicting results could also be due to the different TREM2-deficient mouse models used in each of these studies.

Multiple studies have demonstrated that microglia can actively phagocytize tau, supporting a preponderant role of microglia in the spread of pathological tau throughout the brain [23,34,35,36]. Interestingly, it has been shown how microglia could promote tau-seed propagation due to its failure to process tau aggregates into a non-toxic form, subsequently releasing this tau-seed within exosomes in adjacent cells [23]. Other studies support the idea that microglia secrete factors in addition to tau or tau-containing exosomes that could exacerbate tau spreading. For instance, cytokines released by microglia trigger specific intraneuronal signaling that causes tau seeding and hyperphosphorylation [37,38]. In a groundbreaking study, Asai and collaborators demonstrated that microglial depletion and exosome synthesis inhibition reduce tau propagation in vivo [34], supporting the notion that microglial loss-of-function could be protective in the context of tau pathology spreading.

The role of TREM2 on tau propagation has mainly been studied in the context of amyloid deposition. Genetically decreasing TREM2 levels or altering its function in transgenic mice with amyloid plaque deposits injected with human AD brain-derived tau aggregates enhances tau propagation [19,30,31,32,39,40]. Another study demonstrated that the total removal of microglia enhanced amyloid-associated tau seeding [39], unlike previous observations in an amyloid-free scenario [34]. Altogether, these studies support the notion that, in the context of amyloid plaques, TREM2 function in microglia inhibits amyloid-induced tau propagation. Nevertheless, a very recent study adds some complexity to this assumption. In this novel study, the authors tested the effect of an anti-mouse TREM2 agonist antibody, AL002a, in a combined Ab-amyloid and tau model, hypothesizing that TREM2 activation would inhibit amyloid-induced tau propagation [41]. Unexpectedly, the authors observed an increase in tau spreading in mice treated with the TREM2 antibody compared with that in mice treated with the IgG control [41]. While the authors did not explore why the TREM2 agonist and TREM2 genetic deletion had the same exacerbating effect on tau spreading, one suggested that the explanation lies in the timing of the two models of TREM2 manipulation [42]. In the case of the genetic deletion of TREM2, TREM2 deficiency occurs long before amyloid deposition [19]; however, in the case of TREM2 agonist treatment, the anti-TREM2 antibody was administered at 6 months of age, long after the onset of amyloid pathology [41]. Thus, the effect of TREM2 on tau propagation in an amyloid-dependent manner could depend on the disease stage.

While removing microglia inhibited the spread of tau pathology in an amyloid-free setting [34], a recent study using the adeno-associated virus (AAV) expression of P301L tau showed that TREM2 deletion enhanced tau spreading from the medial entorhinal cortex (MEC) to the hippocampal dentate gyrus (DG) [43]. The authors demonstrated that TREM2 deletion aggravates tau spreading, possibly through microglial extracellular exosomes. Nevertheless, the mechanism by which TREM2 influences the microglial extrusion of tau in exosomes remains unknown. Zhu et al.’s results differ from our observations. These different outcomes could be due to multiple factors. One possible reason is that the effect of TREM2 on tau propagation depends on the tau pathology stage or the tau seed’s nature. In our study, we directly injected mice with tau-PHF with strong seeding activity from AD human brain samples, while Zhu et al. injected AAV-P301L tau to induce the neuronal expression of mutant tau. This model does not form tau PHF-like pathology. Another possible reason is that the effect of TREM2 on tau propagation depends on the disease stage. In our study, the analysis of tau propagation was performed on 9-month-old mice six months post-tau injection. Zhu et al. injected AAV-P301L tau into 4-month-old mice and analyzed them 5 weeks post-injection, representing an acute response to tau propagation.

Trem2-mediated signaling is known to inhibit the transition of microglia to a more Disease-Associated Microglia (DAM) state characterized by enhanced migration, chemotaxis, and phagocytosis [44,45]. Our partial transcriptomic analysis did not reveal any increase in known DAM markers [46] in WT mice injected with AD-tau in comparison with the levels in Trem2^−/−^ mice injected with AD-tau (Appendix A), suggesting that the effect of TREM2 ablation on tau propagation in this model is not due to an effect on the ability of microglia to adopt a DAM phenotype. TREM2-deficient microglia have also been shown to have an impaired proliferative ability and decreased viability [11,47]. Despite the compelling evidence suggesting that TREM2 can regulate microglia phagocytic activity [44,45], in vitro experiments suggest that microglia or macrophage uptake of tau does not depend on TREM2 expression [39,43]. Gratuze et al. found no difference in the uptake of tau-PHF isolated from AD human brains between bone marrow-derived macrophages (BMDMs) cultured from Trem2 wild type (WT) or Trem2 KO mice. However, tau-PHF degradation was delayed in Trem2 KO BMDMs compared with that in Trem2 WT BMDMs [39]. Zhu et al. observed no differences in the uptake of recombinant tau oligomers by WT and Trem2 KO microglia [43]. However, they suggested that TREM2 deletion promotes tau trafficking to late endosome/pre-exosomal compartments [43]. Overall, both studies support the notion that, in the context of tauopathies, microglial dysfunction due to TREM2 downregulation affects tau intracellular trafficking and degradation rather than internalization. In an in vivo setting, the ablation of TREM2 in the PS19 mouse model decreased the number of IBA1-positive microglia in regions affected by tau pathology [13], similar to our observations in the tau propagation model using IBA1 as a microglia marker. The authors did not observe differences in microglia proliferation in PS19 mice regardless of Trem2 genotype, suggesting that TREM2-deficient microglia may undergo metabolic stress, impacting their function and capacity to respond to damage incited by tauopathy, possibly leading to inadvertent cell death [13]. In the same study, a decrease in inflammatory transcripts was observed in PS19 mice due to TREM2 ablation, suggesting that loss of TREM2 function impacts microglial activation, hindering inflammatory responses [13]. Likewise, our partial transcriptomic analysis using NanoString revealed that Trem2^−/−^ mice injected with AD-tau maintain the expression of genes previously identified as contributors to the detrimental effects of tau or which are involved in microglial activation, similar to control mice. These findings suggest that TREM2 deficiency impairs inflammatory responses. Overall, these studies and our data support the notion that reduced TREM2 signaling could render microglia vulnerable to pathological tau aggregates, enhancing their dysfunction, reducing the inflammatory response, and possibly inducing cell death, ultimately decreasing the ability of microglia to contribute to tau propagation.

Our data also suggest that TREM2 deficiency could influence astrocyte responses to tau propagation. Similarly, it has been shown that TREM2 downregulation affects the phenotype of GFAP-positive astrocytes in the APP/PS1 amyloid mouse model and the PS19 model of tauopathy [8,13]. Interestingly, when astrocyte changes were evaluated in the APP/PS1 model, a decrease in GFAP-positive astrocytes but not S100β-positive astrocytes was observed, suggesting that TREM2 deficiency likely affects astrocyte activation rather than the distribution in the brains of mice with amyloid plaques [8]. Considering the heterogeneity of astrocytes in the brain [48], it is also reasonable to suggest that TREM2 deficiency could solely affect one subset of astrocytes. From the aforementioned published studies and our own data, it is still not possible to determine if the effect of TREM2 deficiency on astrocytes in AD models is secondary to the effect of TREM2 on amyloid or tau pathology that subsequently affects the astrocytic response or if TREM2 deficiency has a non-cell-autonomous effect on astrocyte responses. Supporting this last point, a recent study demonstrated that TREM2 is required for microglia to limit astrocyte synaptic engulfment during development [49]. Further studies are necessary to determine if this non-cell-autonomous role of TREM2 could influence astrocyte responses in the context of AD and tau spreading.

## 5. Conclusions

Despite the numerous studies in this area, it remains unclear if the effect of TREM2 deficiency on tau propagation is due to the loss of TREM2 itself as a receptor or whether TREM2 deficiency results in a significant loss of microglial function. Therefore, a better understanding of the physiological role of TREM2 in microglia regulation and how TREM2 signaling affects tau aggregation and propagation in an amyloid-dependent and independent manner is required before targeting TREM2 as a therapeutic entry point for AD.

## Figures and Tables

**Figure 1 cells-12-01597-f001:**
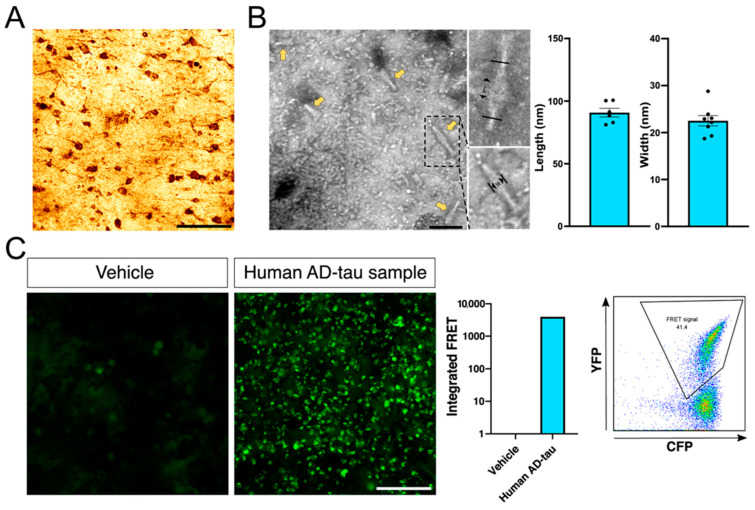
Preparation and characterization of purified human PHF (AD-tau): (**A**) AT8 staining (monoclonal antibody specific for tau phosphorylation at S202/T205) in the temporal cortex tissue section from the AD sample used for tau PHF purification. Scale bar: 100 μm. (**B**) Electron microscopy images of tau fibrils (yellow arrows) and characterization (length and width) from AD extracts. Scale bar: 100 nm (**C**) HEK293T biosensor cells transfected with AD-tau show tau aggregation and increased integrated FRET compared with that in cells transfected with the vehicle alone. Scale bar: 100 μm.

**Figure 2 cells-12-01597-f002:**
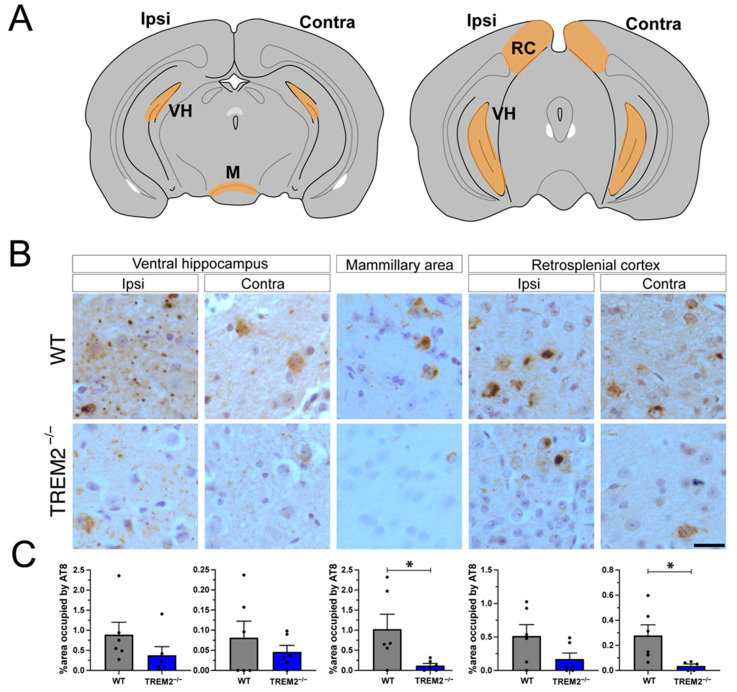
TREM2 depletion ameliorates tau pathology propagation 6 months post-injection: (**A**) Schematic representation of mouse coronal brain sections with the analyzed brain areas highlighted in orange; ventral hippocampus (VH), mammillary area (M), and retrosplenial cortex (RC). (**B**) Immunohistochemistry of AT8 in the previously selected areas in control and Trem2^−/−^ mice. Scale bar: 25 μm. (**C**) Quantification of AT8 staining. Data represent the mean ± S.E.M. Significance was determined using unpaired *t* test (*, *p* < 0.05). *n* = 6 per group.

**Figure 3 cells-12-01597-f003:**
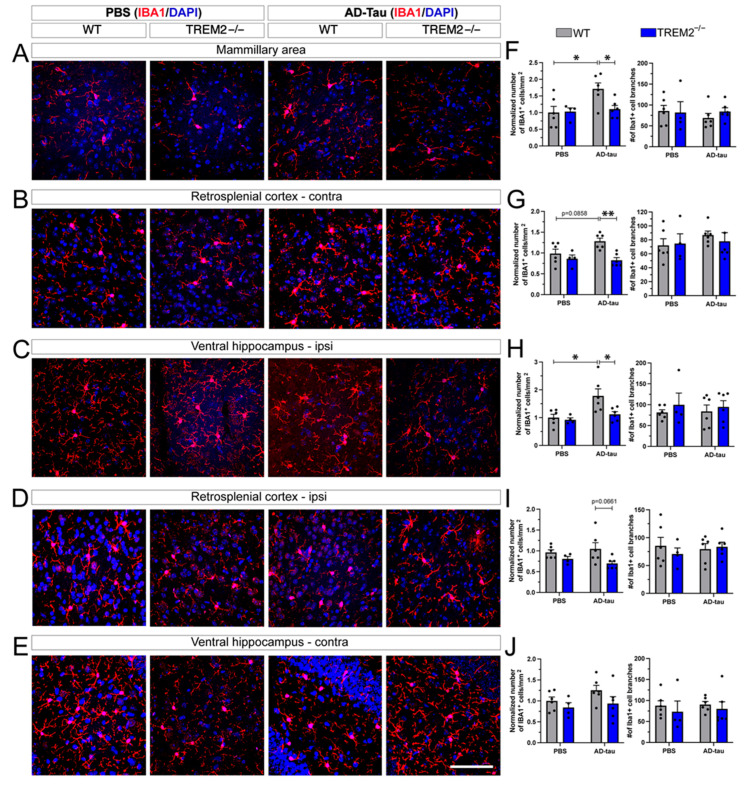
Microglia assessment in Trem2^−/−^ and WT mice injected with AD-tau: (**A**–**E**) IBA1 immunofluorescence in brain regions of mice injected with PBS or AD-tau. (**F**–**J**) Quantification of IBA1^+^ cells/mm^2^ and number of branches in all four groups. Data represent the mean ± S.E.M. Significance was determined using two-way ANOVA (*, *p* < 0.05; **, *p* < 0.01). *n* = 4–6 per group. Scale bar: 50 μm.

**Figure 4 cells-12-01597-f004:**
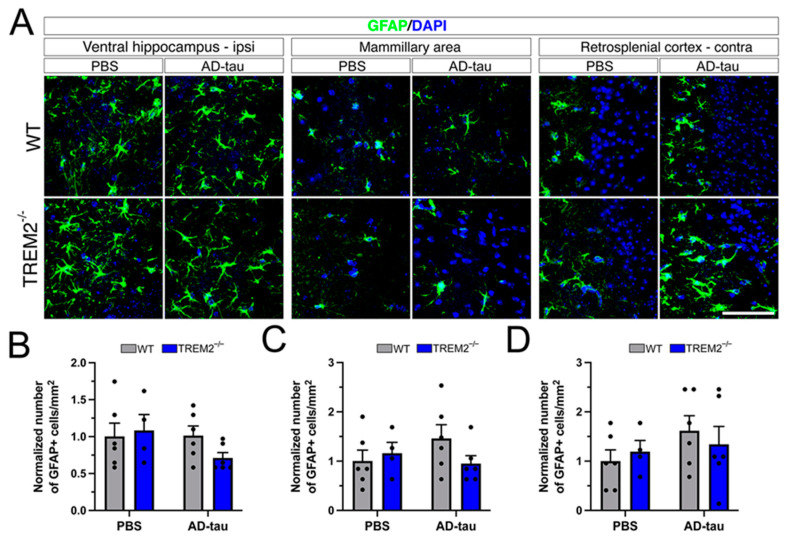
Astrogliosis assessment in Trem2^−/−^ and WT mice injected with AD-tau: (**A**) Immunofluorescence of GFAP in brain regions after PBS or AD-tau injection. (**B**–**D**) Quantification of GFAP^+^ cells/mm^2^ shows no difference across groups. Data represent the mean ± S.E.M. Significance was determined using two-way ANOVA. *n* = 4–6 per group. Scale bar: 50 μm.

**Figure 5 cells-12-01597-f005:**
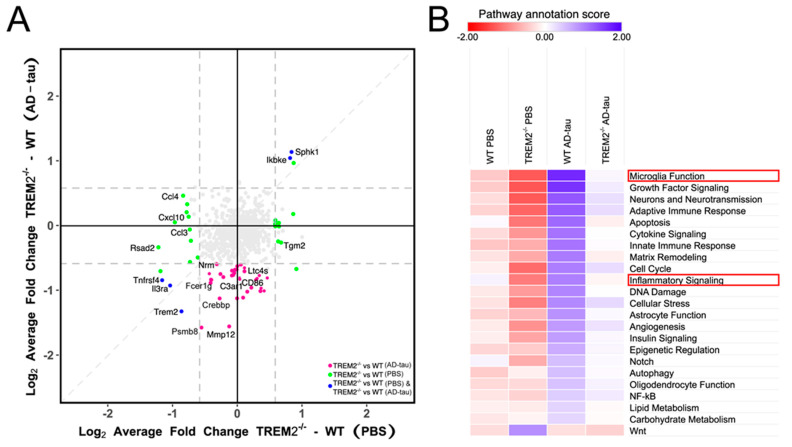
TREM2 depletion modulates neuroinflammatory pathways in response to tau pathology: (**A**) “Four-way” plot comparing differential expression analyses from the NanoString neuroinflammation data in the ipsilateral brain. Each point represents one gene. The *x*-axis shows Log_2_ fold changes in Trem2^−/−^ compared with that in WT controls in PBS-injected mice, and the *y*-axis compares Trem2^−/−^ vs. WT controls in AD-tau-injected mice. Differentially expressed genes (DEGs) with a *p* < 0.05 and log_2_ fold change > ±0.58 from the *x*-axis comparison are shown in green, the DEGs from the *y*-axis comparison are shown in pink, and the DEGs in both comparisons are shown in blue. Gray dots represent genes that were not significantly differentially expressed. (**B**) Comparisons of pathway annotation scores from the NanoString neuroinflammation panel. Microglial function scores and inflammatory signaling differed between WT and Trem2^−/−^ mice injected with AD-tau (highlighted in red). *n* = 6 per group.

**Figure 6 cells-12-01597-f006:**
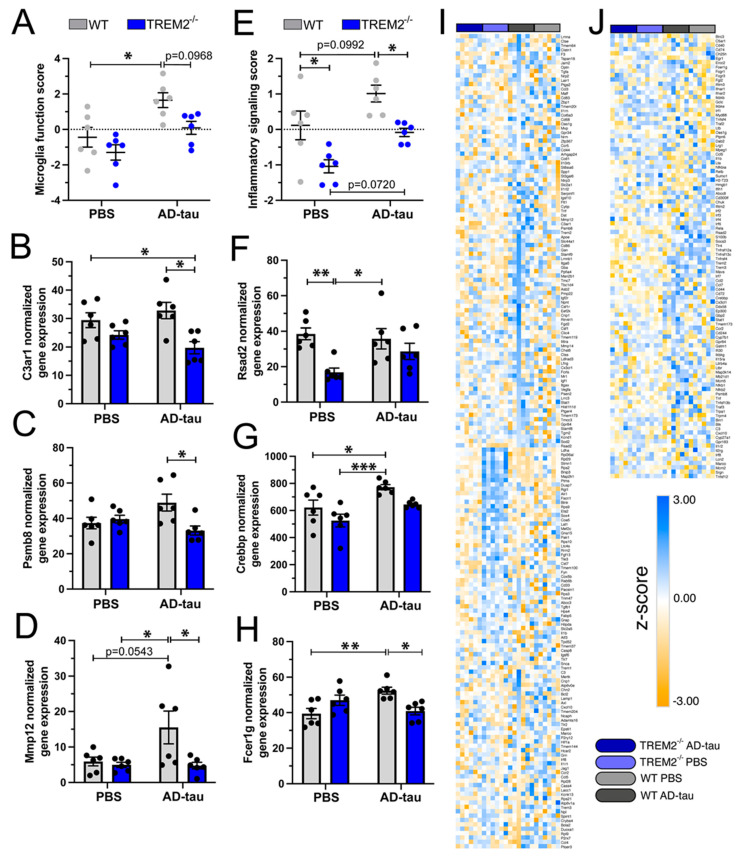
TREM2 depletion reduces microglial function and inflammatory signaling in response to tau pathology. Signaling score of microglial function (**A**) and inflammatory signaling scores (**E**) in Trem2^−/−^ and WT mice injected with PBS or AD-tau. Selected genes from the microglial function (**B**–**D**) and inflammatory signaling pathway annotation scores (**F**–**H**) showing a significant difference between WT and Trem2^−/−^ mice injected with AD-tau. (**I**,**J**) Heat map showing the z-score of all genes included in the microglial function (**I**) and inflammatory signaling (**J**) scores for all groups. Data represent the mean ± S.E.M. Significance was determined using two-way ANOVA (*, *p* < 0.05; **, *p* < 0.01; *** *p* <0.001). *n* = 6 per group.

## Data Availability

All data are available from the corresponding authors upon request.

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
