# Peer review of "TREM2-Deficient Microglia Attenuate Tau Spreading In Vivo"

_cells, 2023, doi:10.3390/cells12121597_

Round 1
Reviewer 1 Report
In this study the authors investigated the effects of TREM2 deficiency on tau spreading using a mouse model in which endogenous tau is seeded to produce AD-like tau features and found that TREM2 deletion may be beneficial in diminishing tau propagation in a pure tauopathy setting. Some concerns and suggestions were listed as below:
When tau pathology propagation can be noted after injection?
Changes of behavior tests of these mice should be measured.
Apart from Iba1, other microglial markers (for example, Tmem119, P2ry12) should also be used.
qPCR should be performed in order to show the changes of gene expression in different groups. This is a major concern.
Soluble and insoluble tau phosphorylation in TREM2 deficient mice should be tested.
Skeletonized microglial data should be provided.
It should be noted that investigators of TREM2 deficiency have employed several different varieties of TREM2 mouse strains which may result in variabilities.
Did you observe Tau-mediated neurodegeneration in this mice model?
Other disease time points should be tested.
Why reduced astrocytosis was not noted in this condition? This point should be discussed.
fine
Author Response
Reviewer 1
In this study the authors investigated the effects of TREM2 deficiency on tau spreading using a mouse model in which endogenous tau is seeded to produce AD-like tau features and found that TREM2 deletion may be beneficial in diminishing tau propagation in a pure tauopathy setting. Some concerns and suggestions were listed as below:
When tau pathology propagation can be noted after injection?
In the original study in which this model of tau propagation was developed (PMID:27810929), in C57/B6 WT mice, tau pathology propagation was observed as early as 3 months post-injection. Nevertheless, tau propagation was more consistent 6 months post-injection. Therefore, we decided to evaluate tau propagation in TREM2 KO mice 6 months post-injection.
Changes of behavior tests of these mice should be measured.
We thank the reviewer for this comment. This AD-tau injection mouse model was originally developed to characterize tau propagation, and no behavioral abnormalities were measured (PMID:27810929). Therefore, to be consistent with previous studies using this in vivo propagation model, we did not evaluate behavioral deficits associated with AD-tau injections in WT or TREM2 KO mice.
Apart from Iba1, other microglial markers (for example, Tmem119, P2ry12) should also be used.
We thank the reviewer for this observation. Considering that IBA1 is used as the marker to quantify changes in microglia in every published study evaluating effect of TREM2 downregulation on tau pathology, we decided to use the same marker for consistency. To address the reviewer’s comments, we have clarified in the discussion section that we quantified Iba1-positive microglia as previously described. Please refer to line 437 to 439.
qPCR should be performed in order to show the changes of gene expression in different groups. This is a major concern.
We thank the reviewer for this comment. The Nanostring platform used in this study quantifies the gene expression (mRNA) levels of over 700 genes. Figure 6 shows the gene expression of selected genes (C3ar1, Psmb8, Mmp12, Rsad2, Crebbp, and Fcer1g) in the four groups. We have provided a Supplementary excel (Supplementary Excel 1) showing the gene expression changes of all the genes for all groups evaluated by Nanostring.
Soluble and insoluble tau phosphorylation in TREM2 deficient mice should be tested.
We thank the reviewer for this comment. This AD-tau injection mouse model was originally developed to characterize tau propagation throughout the brain using histological approaches, and no biochemical changes in tau were determined (PMID:27810929). Therefore, to be consistent with previous studies using this in vivo propagation model, we evaluated tau propagation using the previously described histological approach (PMID:27810929).
Skeletonized microglial data should be provided.
As the reviewer suggested, we have provided skeletonized microglia data. Please refer to Supplementary figure 1 and line 276.
It should be noted that investigators of TREM2 deficiency have employed several different varieties of TREM2 mouse strains which may result in variabilities.
We thank the reviewer for this comment. As the reviewer suggested, in the discussion section, we have included a statement about how the variability observed in the effect of TREM2 deficiency on tau pathology in mice could be due to the different varieties of TREM2 KO mouse strains used in these studies. Please refer to lines 372 to 373.
Did you observe Tau-mediated neurodegeneration in this mice model?
No neurodegeneration was observed in this model, as previously reported (PMID:27810929).
Other disease time points should be tested.
We thank the reviewer for this comment. Considering the time required to perform other in vivo time points, unfortunately, it is not possible to include these in the current study. Nevertheless, we have commented in the discussion that the observed results in our and other published studies could depend on the disease time point selected in each case. Please refer to lines 398 to 399 and 414 to 418.
Why reduced astrocytosis was not noted in this condition? This point should be discussed.
As the reviewer suggested, we have included a discussion about the possible effect of TREM2 deficiency on the astrocytic response associated with tau propagation in this revision. Please refer to lines 453 to 468.

Reviewer 2 Report
1. Many studies have revealed a remarkable heterogeneity among astrocytes. Not all astrocytes in vivo express GFAP, or only weakly so. It makes sense for the authors to investigate the GFAP-negative astrocytes number as well.
2. It seems that there is a trend that the number of astrocytes in Trem-/- with AD-tau reduced compared to WT AD-tau mice. Could authors address that?
3. Astrocyte function seemed also altered for the Term-/- group. It might be interesting to dig a bit more into the pathway and potentially come up with a hypothesis of the relation between tau pathology and astrocyte function.
4. How was the amyloid level in the Trem-/- and AD-tau mice?
N/A
Author Response
Reviewer 2
- Many studies have revealed a remarkable heterogeneity among astrocytes. Not all astrocytes in vivo express GFAP, or only weakly so. It makes sense for the authors to investigate the GFAP-negative astrocytes number as well.
We thank the reviewer for this comment. As mentioned in response to reviewer 1, we have included a discussion about the possible effect of TREM2 deficiency on the astrocytic response associated with tau propagation in this revision and how this effect could occur in a subset of astrocytes, considering the remarkable heterogeneity of these glia cells. Please refer to lines 453 to 468.
- It seems that there is a trend that the number of astrocytes in Trem-/- with AD-tau reduced compared to WT AD-tau mice. Could authors address that?
We thank the reviewer for this comment. The trend mentioned by the reviewer appears to occur solely in one brain region, making it difficult to speculate a possible reason. Nevertheless, in the result section, we changed the statement “No differences in the number of GFAP-positive astrocytes” to “No statistically significant differences in the number of GFAP-positive astrocytes”. Please refer to lines 283 to 284.
- Astrocyte function seemed also altered for the Term-/- group. It might be interesting to dig a bit more into the pathway and potentially come up with a hypothesis of the relation between tau pathology and astrocyte function.
We thank the reviewer for this observation. As the reviewer suggested, we have included the data related to the Astrocyte function score obtained from the Nanostring analysis in the results section (Please refer to Supplementary Figure 3 and lines 329 to 343). We have also included a new section in the discussion concerning the possible effect of Trem2 deficiency on astrocytic function. Please refer to lines 453 to 468.
- How was the amyloid level in the Trem-/- and AD-tau mice?
In the present study, we did not use KI or transgenic mice expressing human forms of APP. Therefore, is not possible to see amyloid in any of the models injected with PBS or AD-tau.

Reviewer 3 Report
In the submitted manuscript Lee-Gosselin et al. present a well-designed study which suggests that loss of TREM2 attenuated tau pathology while impairing neuroinflammation and normal microglial response. As the authors describe there are numerous published manuscripts, with conflicting conclusions, that examine TREM2 in the context of tau pathology in different in vitro and in vivo mouse models. A strength of the current manuscript, which increases its impact, is the model system design. The authors choose to examine one aspect, tau spreading, without the context of other brain pathologies. A glaring question that remained following reading the manuscript is if there is a different initial microglia response in TREM2 deficient mice which results in increased tau pathology at the later timepoint 6 months later or if TREM2 deficiency impacts tau spreading pathways. The authors should present data of the initial response or discuss how a different initial response could impact their later timepoint. Also, what remains clear is if it is loss of TREM2 itself as a receptor that is contributing to the observed changes in tau spreading or that the lack of TREM2 results in a significant microglia loss-of-function, for example in their model would they propose a similar impact on tau spreading with microglia ablation? Addition of this to their conclusions, while speculative, would be impactful. Due to the model of a single infusion of tau isolated from AD human brain, there are some important details related to the methods which the authors should more fully describe. Given that proximity of sections to the infusion site could impact the analysis this should be more clearly described including proximate distances for the different stainings, how during the sectioning process the infusion site was defined, and the number of sections analyzed per animal. The authors discuss the importance of the microglia response to the spreading of tau and this argument could be strengthens with correlations of data from Fig2 and 3. In lines 386-396 the authors clearly put their current study in the context of a previous experiment and the discussion would be improved if they did this more throughout. With the ever-growing microglia nomenclature, including DAM microglia mentioned in the manuscript, the authors should present or discuss how their current Nanostring results compare to the transcriptional signatures of these subpopulations. Furthermore, this data set needs further weight in the discussion (currently 4 lines) including how their changes could relate to tau spreading.
Minor points:
Definition of how significance was defined for the Nanostring data in the methods.
What sexes were used for the mouse studies.
If different animals were used for histological versus Nanostring experiments.
How tissue was chosen for RNA isolation and if the dissected tissue contained the brain regions analyzed in the histological sections.
Barpots with individual points.
Author Response
Reviewer 3
In the submitted manuscript Lee-Gosselin et al. present a well-designed study which suggests that loss of TREM2 attenuated tau pathology while impairing neuroinflammation and normal microglial response. As the authors describe there are numerous published manuscripts, with conflicting conclusions, that examine TREM2 in the context of tau pathology in different in vitro and in vivo mouse models. A strength of the current manuscript, which increases its impact, is the model system design. The authors choose to examine one aspect, tau spreading, without the context of other brain pathologies.
A glaring question that remained following reading the manuscript is if there is a different initial microglia response in TREM2 deficient mice which results in increased tau pathology at the later timepoint 6 months later or if TREM2 deficiency impacts tau spreading pathways. The authors should present data of the initial response or discuss how a different initial response could impact their later timepoint.
We thank the reviewer for this observation. As the reviewer suggested, we have discussed the different initial microglia response in Trem2-/- mice that subsequently affects tau propagation in this revision. Please refer to discussion section lines 353 to 359.
Also, what remains clear is if it is loss of TREM2 itself as a receptor that is contributing to the observed changes in tau spreading or that the lack of TREM2 results in a significant microglia loss-of-function, for example in their model would they propose a similar impact on tau spreading with microglia ablation? Addition of this to their conclusions, while speculative, would be impactful.
We thank the reviewer for this comment. As the reviewer suggested, we have mentioned the necessity of understanding the mechanism by which TREM2 loss of function influences tau propagation in the conclusion. Please refer to the conclusion section, lines 471 to 474.
Due to the model of a single infusion of tau isolated from AD human brain, there are some important details related to the methods which the authors should more fully describe. Given that proximity of sections to the infusion site could impact the analysis this should be more clearly described including proximate distances for the different stainings, how during the sectioning process the infusion site was defined, and the number of sections analyzed per animal.
We thank the reviewer for this comment. In the methods sections, we have included more detailed information about how brain sections were selected for staining. Please refer to lines 179 to 185.
The authors discuss the importance of the microglia response to the spreading of tau and this argument could be strengthens with correlations of data from Fig2 and 3.
As the reviewer suggested, we have included a figure correlating the levels of AT8 (Figure 2) with the number of Iba1-positive microglia (Figure 3). Please refer to Supplementary Figure 2 and the results section (Lines 278 to 281).
In lines 386-396 the authors clearly put their current study in the context of a previous experiment and the discussion would be improved if they did this more throughout. With the ever-growing microglia nomenclature, including DAM microglia mentioned in the manuscript, the authors should present or discuss how their current Nanostring results compare to the transcriptional signatures of these subpopulations. Furthermore, this data set needs further weight in the discussion (currently 4 lines) including how their changes could relate to tau spreading.
As the reviewer suggested, in the revised manuscript, we have included a discussion about the possibility that DAM microglia could be related to tau spreading on the basis of the Nanostring results. Please refer to lines 421 to 425.
Minor points:
Definition of how significance was defined for the Nanostring data in the methods.
As the reviewer suggested, we have discussed how significance was defined for the Nanostring data in the methods section. Briefly, differentially expressed genes (DEGs) were those with a p<0.05 and log2 fold change >±0.58. Please refer to lines 227 to 229.
What sexes were used for the mouse studies.
In the current study, we used an equal number of female and male mice. This information has been included in the methods section. Please refer to lines 235 to 236.
If different animals were used for histological versus Nanostring experiments. How tissue was chosen for RNA isolation and if the dissected tissue contained the brain regions analyzed in the histological sections.
As the reviewer requested, this information has been included in the methods section. Briefly, a different cohort of mice was used for the Nanostring analysis (Please refer to line 218). The whole hemisphere was used for this analysis (Please refer to line 220).
Barpots with individual points.
As the reviewer suggested, all figures have been updated to barplots.

Round 2
Reviewer 1 Report
The authors have answered my questions.
fine